# Osimertinib-Associated Toxic Epidermal Necrolysis in a Lung Cancer Patient Harboring an EGFR Mutation—A Case Report and a Review of the Literature

**DOI:** 10.3390/medicina56080403

**Published:** 2020-08-11

**Authors:** Izumi Sato, Hiroki Mizuno, Nobutaka Kataoka, Yusuke Kunimatsu, Yusuke Tachibana, Takumi Sugimoto, Nozomi Tani, Yuri Ogura, Kazuki Hirose, Takayuki Takeda

**Affiliations:** Department of Respiratory Medicine, Japanese Red Cross Kyoto Daini Hospital, Kyoto 602-8026, Japan; izumi-h@koto.kpu-m.ac.jp (I.S.); gx13b46@stu.kpu-m.ac.jp (H.M.); nkataoka@koto.kpu-m.ac.jp (N.K.); ky92020223@yahoo.co.jp (Y.K.); yutachib@koto.kpu-m.ac.jp (Y.T.); ta9ta9@koto.kpu-m.ac.jp (T.S.); nozomi-t@koto.kpu-m.ac.jp (N.T.); yuri.krn.cncl@gmail.com (Y.O.); k-hirose09@outlook.jp (K.H.)

**Keywords:** EGFR tyrosine kinase inhibitor, ethnicity, non-small-cell lung cancer, osimertinib, toxic epidermal necrolysis

## Abstract

Toxic epidermal necrolysis (TEN) and Stevens-Johnson syndrome (SJS) are life-threatening dermatologic adverse events in the same category, caused by a delayed-type drug hypersensitivity reaction. Although skin toxicity is common during treatment with epidermal growth factor receptor tyrosine kinase inhibitors (EGFR-TKIs), osimertinib-associated TEN is quite rare—thus far, only one report has been published from China. We report a case of an 80-year-old Japanese woman with lung adenocarcinoma harboring an EGFR-sensitizing mutation who was treated with osimertinib as the first-line treatment. Forty-six days after osimertinib induction, diffuse erythematous rash rapidly spread over the patient’s trunk along with vesicles and purpuric macules; furthermore, she developed targetoid erythema on the face. Despite osimertinib discontinuation and corticosteroid treatment, diffuse erythema with Nikolsky’s sign, general epidermal detachment, erosion and loose blisters developed over her entire body including the face. Based on her symptoms, TEN was diagnosed and thus, intravenous immunoglobulin was immediately administered for 4 days. The treatment ameliorated TEN-associated skin toxicity and caused epithelialization. Reports on osimertinib-associated SJS/TEN are scarce and only one report each on SJS and TEN from China is available. This is the first report of osimertinib-associated TEN from Japan. Cases of EGFR-TKI-associated SJS/TEN have been reported predominantly from Asian countries, suggesting ethnicity and genetic linkage play a role in the underlying mechanism.

## 1. Introduction

The discovery of epidermal growth factor receptor (EGFR) mutation [1,2] and the introduction of EGFR-tyrosine kinase inhibitors (EGFR-TKIs) have revolutionized the treatment strategy for non-small-cell lung cancer (NSCLC) harboring EGFR-sensitizing mutations [3]. Osimertinib is a third-generation irreversible EGFR-TKI, whose efficacy has been confirmed in patients who showed disease progression despite treatment with first- or second-generation EGFR-TKIs and harbored the EGFR T790M [4], as well as treatment-naïve patients with an EGFR mutation in the first-line setting [5].

Toxic epidermal necrolysis (TEN) is a quite rare but life-threatening dermatologic adverse event (AE), potentially occurring due to a medication; its annual incidence has been estimated to be 1.58 to 2.26/million [6]. Because TEN develops rapidly accompanied by systemic symptoms, its prompt recognition and adequate management are of the utmost importance. TEN and Stevens-Johnson syndrome (SJS) are regarded as the variants of the same clinical entity and are attributable to a delayed-type hypersensitivity reaction to a drug [6]. Previous reports on SJS/TEN associated with osimertinib are scarce, with only one report each on SJS [7] and TEN [8] available from China. We herein report the first Japanese case of osimertinib-associated TEN with a literature review of TEN associated with gefitinib [9] and osimertinib [8] as well as three SJS cases, of which two was associated with afatinib [10,11] and the other with osimertinib [7].

## 2. Case Report

An 80-year-old woman with adenocarcinoma (cT4N0M1c [BRA, PUL], stage VIB) and harboring an EGFR-sensitizing mutation (L858R) was treated with osimertinib in December 2018 as the first-line treatment, which led to a partial response with marked tumor shrinkage, which was assessed by chest computed tomography after 3 weeks of osimertinib treatment in January 2019. Erythematous papules were observed on the anterior chest wall 32 days after osimertinib induction. Therefore, the patient was referred to the department of dermatology of our institution. Since the dermatologic adverse event (AE) was diagnosed as mild at that time, a topical corticosteroid ointment was prescribed without cessation of osimertinib treatment. However, diffuse erythematous rash rapidly spread over the patient’s trunk (Figure 1A) and was accompanied by vesicles and purpuric macules; furthermore, targetoid erythema developed on her face (Figure 1B) after 14 days. Therefore, osimertinib was immediately discontinued. However, the erythema and macules did not alleviate after osimertinib discontinuation and oral prednisolone at a dose of 30 mg was initiated 5 days after osimertinib discontinuation. Four days after the initiation of prednisolone, diffuse erythema with Nikolsky’s sign, general epidermal detachment, erosion and loose blisters developed over her entire body including the face (Figure 1C). Based on these symptoms, a clinical diagnosis of TEN was made, which was supported by a skin biopsy showing acute interface dermatitis with eosinophil infiltration (Figure 1D). Therefore, intravenous immunoglobulin (IVIG; 15 g/day) was immediately administered for 4 days. Ten days after IVIG treatment, the erythematous rash on the patient’s trunk disappeared and some scar lesions remained (Figure 2A). Furthermore, general epidermal detachment, erosion and loose blisters on her face ameliorated with epithelialization (Figure 2B). Therefore, oral prednisolone was gradually tapered off.

## 3. Discussion

Although SJS/TEN is an extremely rare AE associated with any medication including EGFR-TKIs, it requires close attention considering its rapid progression and the possibility of severe ocular, cutaneous and renal complications, which may impair patients’ quality of life. It could also be life-threatening and is associated with a high mortality rate. SJS/TEN usually develops following a prodromal illness resembling an upper respiratory tract infection for several days before the rash appears and symptoms include fever (body temperature, >39 °C), sore throat, rhinorrhea, cough, conjunctivitis and arthralgia. SJS/TEN is categorized into three distinct types based on the degree of skin involvement. Skin detachment of less than 10% is categorized as SJS, 10–30% as SJS/TEN overlap and >30% as TEN [6]. The corresponding mortality rate for each type has been reported to be 4.8%, 19.4% and 14.8% [6].

Drug-specific cytotoxic T lymphocytes activated by a drug attack keratinocytes by direct contact, which induces apoptosis. Furthermore, cytokines generated through the reaction, such as tumor necrosis factor-alpha, perforins and granzymes, could exhibit cytotoxicity. Excessive immunologic reactions due to type 1 helper T (Th1) lymphocytes are suppressed by regulatory T lymphocytes (Tregs) under normal conditions. However, an imbalance could be induced between Tregs and Th1 lymphocytes if a specific drug sensitizes Th1 lymphocytes into drug-specific T lymphocytes. Antibiotics such as sulfonamides, anticonvulsants, oxicam-type non-steroidal anti-inflammatory drugs and allopurinol are high-risk drugs for SJS/TEN [6].

Skin toxicity is one of the most prevalent AEs associated with EGFR-TKI treatment and could have varied manifestations including acneiform rash, dry skin, erythema, fissures and cracks, paronychia and pruritis, which are common and can be well tolerated upon using ointments and bandages [12,13]. Dermatologic AEs occur since EGFR-TKIs inhibit EGFRs, which are highly expressed in skin epithelial cells, leading to mucocutaneous toxicities. The incidence of dermatologic AEs is higher among patients treated with second-generation EGFR-TKIs (such as afatinib and dacomitinib) than among those treated with first-generation EGFR-TKIs (such as gefitinib and erlotinib). This is so because second-generation EGFR-TKIs irreversibly and extensively inhibit pan-human EGFR family signaling. In contrast, third-generation EGFR-TKIs have been designed to inhibit mutated EGFR and are rarely associated with dermatologic AEs. Although ethnic differences in skin toxicities have not been observed in phase III trials of first-, second- and third-generation EGFR-TKIs, Japanese patients tend to develop a rash more frequently compared to their global, European and non-Japanese Asian counterparts [5,13,14]. However, only a few SJS/TEN cases associated with EGFR-TKIs have been reported [7,8,9,10,11]. The rareness of SJS/TEN associated with EGFR-TKIs is due mainly to the difference in the mechanism between usual types of skin toxicity and SJS/TEN. SJS/TEN as the most severe type of dermatologic AEs associated with EGFR-TKIs has been reported predominantly from Asian countries, with afatinib-associated SJS reported from Germany and Japan [10,11], osimertinib-associated SJS from Taiwan [7] and TEN associated with gefitinib [9] and osimertinib [8] from China (Table 1). This is the first report of a Japanese case of osimertinib-associated TEN. All three cases of TEN associated with EGFR-TKIs have been reported from Asian countries, suggesting that ethnicity plays a role in the underlying mechanism.

Ethnicity as an underlying mechanism of SJS has been rigorously investigated in carbamazepine (CBZ)-associated SJS, linked with the first report on a dominant correlation between the *HLA-B*1502* allele and CBZ-associated SJS in the Han Chinese [15]. The prevalence of CBZ-associated SJS in Asians has been reported to account for 25–33% of all SJS cases [16] and is higher than the prevalence among Europeans (5–6%) [17]. Furthermore, only four out of twelve European patients in a European study (RegiSCAR) on CBZ-associated SJS/TEN had the *HLA-B* 1502* allele and all four patients had an Asian ancestry [18], suggesting that ethnicity plays a role in the development of SJS following CBZ administration.

Considering that TEN is the severest form in the SJS/TEN spectrum, ethnicity could be a major factor influencing the development of TEN. Furthermore, the genetic linkage of HLA- and non-HLA genes and SJS/TEN associated with EGFR-TKIs could be an important factor that would explain the underlying mechanism as well as the role of ethnicity in SJS/TEN development and needs further investigation.

## 4. Conclusions

Although SJS/TEN is an extremely rare AE associated with EGFR-TKIs, it requires close attention considering its rapid progression, possibility of severe complications and a high mortality rate. SJS/TEN associated with EGFR-TKIs has been reported predominantly from Asian countries, ethnicity could be a major factor influencing its development.

## Figures and Tables

**Figure 1 medicina-56-00403-f001:**
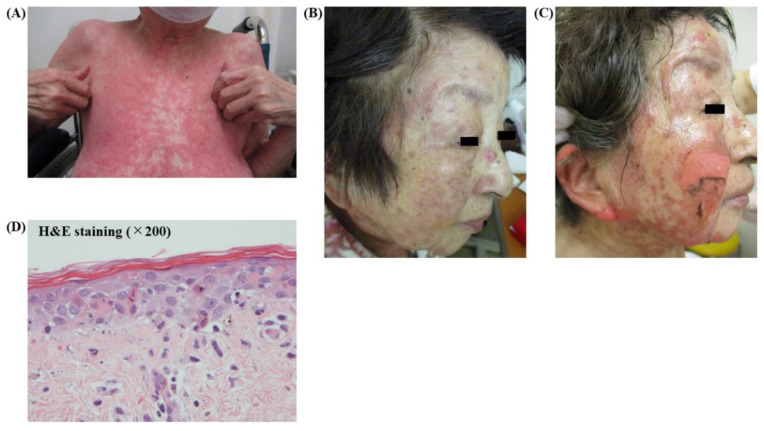
Diffuse erythema spread over the trunk (**A**) and targetoid erythema multiforme on the face (**B**) were observed 46 days after osimertinib induction. General epidermal detachment, erosion and blisters on the entire body, including the face, were observed on day 55 (**C**). A skin biopsy on day 55 showed extensive liquefaction degeneration in the area of the dermal-epidermal junction with sparse inflammatory infiltration of eosinophils (**D**).

**Figure 2 medicina-56-00403-f002:**
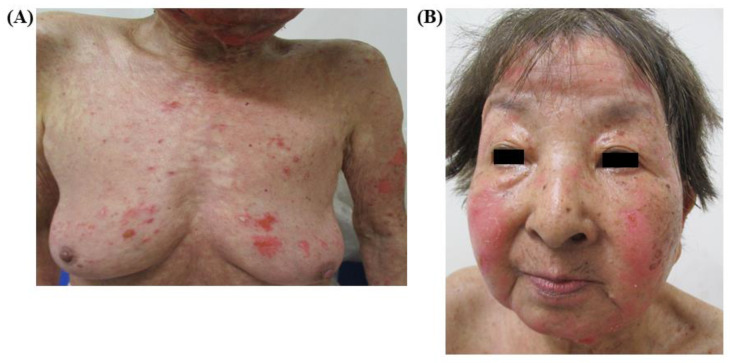
Diffuse erythema on the trunk (**A**) as well as erosion and blisters on the face (**B**) greatly improved 10 days after intravenous immunoglobulin therapy. The patient selected best supportive care thereafter, since the response was maintained after the cessation of osimertinib.

**Table 1 medicina-56-00403-t001:** Stevens-Johnson syndrome and toxic epidermal necrolysis induced by EGFR-tyrosine kinase inhibitors (EGFR-TKIs): characteristics of the current case and previously reported cases.

Author, Reported Year	Age	Sex	Nationality	EGFR-TKI	SJS/TEN	Onset	Treatment and Outcome
Huang, et al., 2015. [9]	42	female	Chinese	gefitinib	TEN	8 days	mPSL and IVIG, recovery after 40 days
Doesch, et al., 2016. [10]	79	female	German	afatinib	SJS	64 days	PSL, recovery after 60 days
Otsuka, et al., 2016. [11]	65	female	Japanese	afatinib	SJS	32 days	mPSL and IVIG, recovery after 60 days
Wang et al., 2018. [8]	51	male	Chinese	osimertinib	TEN	21 days	mPSL and IVIG, recovery after 30 days
Lin et al., 2019. [7]	57	female	Taiwanese	osimertinib	SJS	22 days	PSL, recovery after 60 days
Current case. 2020.	80	female	Japanese	osimertinib	TEN	32 days	PSL and IVIG, recovery after 80 days

Abbreviations: EGFR-TKI, Epidermal growth factor receptor tyrosine kinase inhibitor; IVIG, intravenous immunoglobulin; PSL, prednisolone; mPSL, methylprednisolone; SJS, Stevens-Johnson syndrome; TEN, toxic epidermal necrolysis.

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
