# Peer review of "Osimertinib-Associated Toxic Epidermal Necrolysis in a Lung Cancer Patient Harboring an EGFR Mutation—A Case Report and a Review of the Literature"

_medicina, 2020, doi:10.3390/medicina56080403_

Round 1

Reviewer 1 Report

This case report reported osimertinib-associated epidermal toxicity. In general is interesting know about side effects of different treatment.

My question is about why these consequenses are so rare and they are statistically significant compared to the all population treated with osimertinib? It ethnicity specific?

Author Response

I appreciate your sincere suggestion.

As stated in the Introduction and Discussion parts, toxic epidermal necrolysis (TEN) is a quite rare but life-threatening dermatologic adverse event. Furthermore, the reports on epidermal growth factor receptor-tyrosine kinase inhibitors (EGFR-TKIs) are extremely rare which are reviewed in Table 1, and osimertinib-associated TEN has only one report available from China.

Then, I think the current first case in Japan is quite rare, which is the second osimertinib-induced TEN report in the world.

 In table 1, we have reviewed literatures on EGFR-TKIs-induced TEN and Stevens-Johnson syndrome (SJS), which is regarded as the variants of the same clinical entity. Through the review, we noticed the ethnicity as a tentative reason why EGFR-TKI-induced SJS and TEN had been predominantly reported from Asian country.

Reviewer 2 Report

Dr.Sato and coll. presented an interesting case report regarding toxic epidermal necrolysis in an 80-years old woman treated with Osimertinib for a metastatic, EGFR mutated lung adenocarcinoma.
- The introduction is clear and comprehensive, with appropriate references.
- For what concerns the presentation of the case report, I suggest adding the date of diagnosis and the date of the first disease revaluation (the Authors reported a " partial response with marked tumor shrinkage", but it is not specified how and when the disease assessment was made).
It is not clear what the therapeutic approach was after the resolution of skin toxicity. Has the patient undergone other treatments for her lung cancer?
- Discussion and conclusions are well structured and clear.
I believe that this case report could be adequate for publication in the Medicina, with minor revisions and English spell-check.

Author Response

I really appreciate your sincere advice and comments to improve our manuscript.

I added the relevant data on the osimertinib introduction and evaluation, in which I omitted the exact date in order to ensure the anonymity of the current case.

And the patient selected best supportive care, which was added in the Case Report part.

The manuscript has already been checked by a native speaker (Editage incorporated). I am sorry if the English level has not reached the level needed for publication in Medicina.

I am willing to ask the company to issue the certificate, if needed.